# Self-Reporting of Teacher–Student Performance in Virtual Class Interactions in Biological Sciences during the SARS-CoV-2/COVID-19 Pandemic

Aldo Bazán-Ramírez [1,*], Homero Ango-Aguilar [2], Víctor Cárdenas-López [2], Roberta B. Anaya-González [2], Walter Capa-Luque [3] and Miguel A. Bazán-Ramírez [4]

1 Escuela de Medicina, Universidad César Vallejo, Trujillo 13007, Peru
2 Facultad de Ciencias Biológicas, Universidad Nacional de San Cristóbal de Huamanga, Ayacucho 05001, Peru; homero.ango@unsch.edu.pe (H.A.-A.); victor.cardenas@unsch.edu.pe (V.C.-L.); roberta.anaya@unsch.edu.pe (R.B.A.-G.)
3 Facultad de Psicología, Universidad Nacional Federico Villarreal, Lima 15082, Peru; wcapa@unfv.edu.pe
4 Escuela Profesional de Ingeniería Agroindustrial, Universidad Nacional Federico Villarreal, Lima 15082, Peru; mbazancrist@hotmail.com
* Correspondence: abazanramirez@gmail.com

**Abstract:** We used an interbehavioral model of teacher–student didactic performance with six pairs of criteria, as follows. Exploration of competencies and precurrent for learning, explicitness of teacher criteria and identification of student criteria, teacher illustration and illustration—student participation, supervision of the practice by the teacher and pertinent student practice, teacher feedback and feedback—student improvement, and teacher evaluation and evaluation—student application. The objective was to determine the level of covariation, divergence and convergence between the constructs of teacher didactic performance and student didactic performance in virtual classes as a result of the SARS-CoV-2 pandemic, in November 2020. Three hundred-thirty undergraduate students of biological sciences from a public university in Peru completed two self-report scales in virtual modality through Google forms, one on the performance of their teachers and the other on their own performance. By means of confirmatory factor analysis and an analysis of the covariance of teacher–student performance constructs, three models were obtained with good fits for the convergent and divergent validities of six constructs of the two teaching performance scales. Adequate models of functional correspondence in each pair of the six teacher–student didactic performance criteria were obtained. Likewise, didactic performances related to the identification of teaching–learning criteria and to the evaluation—application criteria were the most frequent during teaching–learning, according to the students' self-report.

**Keywords:** teaching; student learning; higher education; teacher's competences; academic improvement

## 1. Introduction

Interest in the study of interactions in teaching and learning in higher education includes the competencies of teachers and students when they interact in face-to-face or virtual didactic situations, theoretical, theoretical–practical or in laboratory learning [1–3]. The creation and implementation of didactic experiences has focused on interactions that benefit the teaching–learning process and the sustainable development of university students. This has been achieved through the acquisition of competencies and skills that are in accordance with the expected learning and with the criteria established for the achievement of such expected learning [4,5].

Educational processes, and in particular the experiences of didactic teacher–student interactions, also help people to acquire knowledge, skills, values and behaviors related to sustainable development in their daily life and work. These competencies and experiences, of both teachers and students, can be facilitated through various psychological and

pedagogical approaches and didactic interaction practices, in order to develop learning in various disciplines [6–9].

Some authors have pointed out that competency-centered teaching models at universities are based on didactic principles of professional direction, interdisciplinarity, grounding and informatization [1]. It has also been reported that, in order to achieve didactic objectives, teachers focus on active learning activities that foster responsibility and ethics, enhance learning and favor the development of competencies [2]. Likewise, some models that teachers use at university, including traditional, technological, spontaneous or active, and constructivist or alternative–investigative, have been characterized [3]. For example, in a study of students of educational and technological sciences at an Ecuadorian university, it was reported that the technological and spontaneous–active models used by teachers were the best predictors of academic achievement [4].

Regardless of the teaching–learning models and the differences in the models of analysis of the didactic interactions in the classroom, either face-to-face or virtual, teachers and students deploy different actions when interacting in teaching and learning situations [5].

The teacher–student interaction is essential for learning. During these interactions the teacher directs, manages and facilitates the learning of their students through different performances [6,7]. The students' performances during the interactive learning processes, and after them, functionally correspond to the teacher's performances in this didactic interaction [8,9]. Additionally, teachers act as facilitators and motivators for students to learn in online classes [10].

Before the virtualization of classes across the world in the university context due to the SARS-CoV-19 pandemic, there was already great interest in the study of teacher and student performance during didactic interactions [11–14]. Research on the subject has shown that various aspects of the didactic interaction in face-to-face classes have affected the quality of instruction, the quality of learning and the academic achievement of students and has shown a low efficiency in terms of classroom interaction and an inefficiency in terms of the feedback of classroom interaction [9,11–14]. On the other hand, it has been reported that classes in smart classrooms have a better effect on teacher–student interaction in contrast with classes in traditional multimedia classrooms [15].

### 1.1. The Virtualization of Teaching at the Level of Higher Education during the Pandemic

The advent of virtual classes in universities worldwide due to the SARS-CoV-2 pandemic has meant that educational institutions which offered face-to-face programs have had to transform their entire educational offering to the virtual modality [16–23]. Most university teachers adapted to an asynchronous virtual work dynamic [16], and these innovations generated a new online space and better opportunities for didactic interactions and university learning [17]. Even laboratory sessions in various disciplines were replaced by interactive online practices and workshops [18–20], especially in areas related to biological sciences [21–23].

The new reality of teaching–learning during the pandemic has also motivated didactic interactions and virtual practical classes to become scenarios for the study of the didactic performances of teachers and students [24]. Thus, the characterization of teacher–student didactic performances in face-to-face classes can be adapted to identify teacher–student performances in didactic interactions in virtual environments [25,26]. However, teaching in virtual environments has its own difficulties. For example, it has been reported that the effect of the transfer of digital competencies (information acquisition, communication, self-assessment, access to information, the application of digital security and problem solving), depends on the types of interaction with respect to the level or degree of studies and age groups [27]. On the other hand, discrepancies have been reported between the teacher's perception and the university student's perception of teacher performance and teaching quality [28,29], which are similar in the face-to-face context [7]. Additionally, a digital divide between faculty and students, low digital literacy, and inequality of access and connectivity have been reported [17,30].

Among other aspects that have been investigated; synchronous virtual teaching–learning interactions seem to have a greater effect on learning experiences and outcomes, in contrast with asynchronous-type interactions, especially in the didactic performance of feedback and collaborative learning [31]. However, it is necessary to consider more didactic performance variables, which constitute the core of pedagogical interaction in online and offline environments, and include performance, motivation, communication, cognition, emotions, physical state, and temporal dimension [32]. Another aspect investigated in virtual teacher–student interactions was the self-perception of teachers regarding the different professional and interpersonal performances of teachers and students. Here, variables such as accessibility, attention, support and trust stand out as important aspects of the quality of interactions between teachers and students in higher education [28].

It has also been reported that there are no significant differences in the competency development achieved by students in didactic interactions in practices under a blended learning (b-learning) educational model versus a face-to-face teaching system model [33]. Similarly, it has been reported that students in the new remote university campuses feel challenged by the learning environment, the number and quality of teachers, the learning interaction spaces, the virtual campus and urban environment, and the university and community culture [34].

### 1.2. The Research Problem

In the context of virtual teaching during the pandemic and post-pandemic in educational programs that were historically taught face-to-face, it is necessary to conduct studies to describe the didactic performance of both teachers and students in theoretical or practical and laboratory classes. Unfortunately, few studies have been reported on the performance or competencies of teachers and students interacting in classes or practices in terms of the adaptation to a virtual environment that was due to the SARS-CoV-2 pandemic in the university context. This is the case regardless of the methodology and study techniques, whether they are analyses with observational records, or whether they are self-reports and/or interviews.

Our research has focused on interactions in both theoretical and practical classes, in a university undergraduate program in biological sciences in the context of the SARS-CoV-2 pandemic, with the use of self-reports by the students on the teaching performance of their teacher and the students themselves.

For the research reported in this article, the following questions were posed, in order to analyze perceptions of these interactions, with self-report scales on the following didactic performance criteria: 1. What is the level of covariation and divergence and convergence between constructs of didactic performance of the teacher and the didactic performance of the student in virtual classes? 2. Is there convergent and divergent validity in each of six pairs of didactic performance criteria for teacher–student interactions? 3. What is the student's perception of the criteria determining teacher performance and student performance in virtual classes?

### 1.3. The Conceptual Framework of This Study

Educational events, such as didactic interaction processes, constitute fields of psychological interaction [35,36], and every psychological event constitutes an integrated field of relationships [37]. Interactions between the teacher, the learner, and events or topics on what is taught and on what is learned can be structured based on various hierarchically organized functional levels [38]. Based on the principles of the interbehavioral psychology of J.R. Kantor [35,36], an interbehavioral model of didactic performance has been proposed for the study of interactions during classes or practices in science teaching and learning [9,39,40]. The categories for the student, which should functionally correspond with the categories for the teacher, were outlined theoretically [8] and concretized and validated on the basis of observational records of real situations of didactic interaction [7].

Self-reports derived from this model of didactic performance have achieved stability and construct validity with restricted models of both teacher and student, and of various disciplines, contexts and educational levels [7,29,41]. Likewise, good goodness-of-fit has been found in predictive structural regression models with structural equation modeling. Table 1 shows the six pairs of didactic performance criteria used in this study, based on previous validations [7,29,41], from the initial proposals of the interbehavioral model of didactic performance [39,40].

**Table 1.** Didactic performance criteria for teacher—students.

| Teacher Didactic Performance Criteria | Student Didactic Performance Criteria |
| --- | --- |
| Competence exploration | Precurrents to learning |
| Explicitness of criteria | Identification of criteria |
| Illustration | Illustration—Participation |
| Practice supervision | Pertinent practice |
| Feedback | Feedback—Improvement |
| Evaluation | Evaluation—Application |

*1.4. Objectives*

1.4.1. General Objective

To determine the level of covariation and divergence and convergence between constructs of teacher didactic performance and student didactic performance in virtual classes based on the self-reported assessment of students of the Professional School of Biology of a public university in Peru, in the context of the SARS-CoV-2 pandemic, in the August–November period (Semester 2020-I).

1.4.2. Specific Objectives

1. To obtain convergent and divergent construct validity in each of the six pairs of teacher–student didactic performance criteria.
2. To describe the level of presentation of the students' assessment of each of the criteria of teacher performance and student performance in virtual classes.

**2. Materials and Methods**

*2.1. Research Model*

Basic research was conducted under a quantitative, descriptive approach with structural regression modeling. A non-experimental, cross-sectional design was used with survey techniques (application of self-report scales), also known as instrumental cross-sectional design [42], which collects information with self-reports.

*2.2. Study Variables*

The six variables with which to identify teacher didactic performance and the six student performance criteria variables shown in Table 1 were taken in accordance with the substantive theory that underlies the measurement of didactic performance under the interbehavioral perspective of psychology [6–9,39–41]. As several methodologists have already stated, it is important to have a substantive theory that makes sense and guides both the measurement and validation of constructs in the measurement and evaluation of behaviors [43–45].

The following six pairs of didactic performance variables were included: exploration of competencies and precurrent for learning, explicitness of teacher criteria and identification of student criteria, teacher illustration and illustration—student participation in virtual classes, supervision of the practice by the teacher and pertinent student practice, teacher feedback and feedback—student improvement, and teacher evaluation and evaluation—student application. A complete description of these twelve variables (six pairs of didactic performance criteria) are found in Table 2.

**Table 2.** Teacher and Student Didactic Performance Criteria.

| Teacher Didactic Performance | Student Didactic Performance |
|---|---|
| Competency exploration. Consists of the evaluation of prerequisite knowledge and competencies in the student before starting a course or class by the teacher. | Precurrent learning behaviors. The student demonstrates whether or not he/she possesses prerequisite knowledge and competencies for new learning. |
| Criteria explicitness. Involves the teacher explaining the measure of expected achievements according to the didactic criteria established. | Criteria identification. The student demonstrates knowledge of the didactic criteria to adjust his/her performance and reach the course or class achievements. |
| Illustration: The teacher models and establishes action guidelines through didactic discourse to achieve learning achievements. | Illustration—Participation: The performance and learning achievements attained by the student correspond to the criteria and guidelines established by the teacher. |
| Practice supervision: The teacher regulates and facilitates the student's progressive learning, generating the necessary conditions for achievement according to the expected criteria. | Pertinent practice: The student demonstrates proficient performance that conforms to the requirements and compliance criteria modeled by the teacher in practice. |
| Feedback: The teacher gives feedback to the student on the level of his or her learning with respect to the instructional objectives and expected achievements. | Feedback—Improvement: The student evaluates their learning process based on the teacher's feedback, and thus makes the necessary adjustments to optimize their learning. |
| Assessment: The teacher evaluates the student's performance according to the established didactic criteria and makes pertinent adjustments if necessary. | Assessment—Application: In order to establish the performance parameters according to the established criteria, the student faces new problems and situations. |

*2.3. Hypotheses*

**Hypothesis 1.** *The two scales measuring teacher–student teaching performance have satisfactory convergent and divergent construct validity and good goodness-of-fit with respect to the theoretical models of confirmatory factor analysis with structural equation modeling.*

**Hypothesis 2.** *There is a significant functional relationship between teacher didactic performance and student didactic performance (second-order latent variables) in virtual life science classes, as perceived by students.*

**Hypothesis 3.** *There is convergence and divergence of constructs between each of six pairs of teacher–student didactic performance in virtual life science classes measured with student self-participation.*

**Hypothesis 4.** *The evaluation of the biological sciences student body is similar to the criteria for teacher performance and student performance criteria.*

*2.4. Participants*

The population consisted of 552 students enrolled in the 2020-I semester of the Professional School of Biology of the Faculty of Biological Sciences at the UNSCH. The sample consisted of 330 students enrolled in the 2020-I semester who agreed to participate voluntarily in the present research and signed the respective informed consent form.

*2.5. Research Instrument*

**Student rating scale on teacher performance in virtual classes.** To identify the criteria (variables) of the didactic performance of the teacher in didactic interactions in classes and practices in the virtual modality, perceived by students of biological sciences of a

public university in Peru, the scale of Teaching Performance of the Psychology Professor was adapted. This scale has been validated for the evaluation of teaching performance in undergraduate psychology professors in the face-to-face modality [41] and has been extended and validated for the evaluation of the didactic performance of graduate professors in educational sciences in the virtual modality that was adopted due to the SARS-CoV-2 pandemic [29]. This instrument consists of 24 items (Appendix A), four items for each of the six criteria of teacher performance, according to the interbehavioral model of didactic performance and the substantive theory from which they were derived [35–40]. Each student response was rated on a Likert scale, 0 = never, 1 = sometimes, 2 = almost always, and 3 = always.

Figure 1 presents the factor structure of the Student Rating scale on teacher performance in virtual classes. The goodness-of-fit indices obtained by means of AFC using WLSMV as estimator, allow us to argue that the multidimensional model presents satisfactory evidence of validity based on the internal structure of the construct. The global fit results are all optimal: $\chi^2$ (237) = 477.738, CFI = 0.977, TLI = 0.973, RMSEA = 0.055 (0.048, 0.063), SRMR = 0.044. As for the relationships between the factors and the items, they are all greater than 0.65, i.e., high standardized factor loadings.

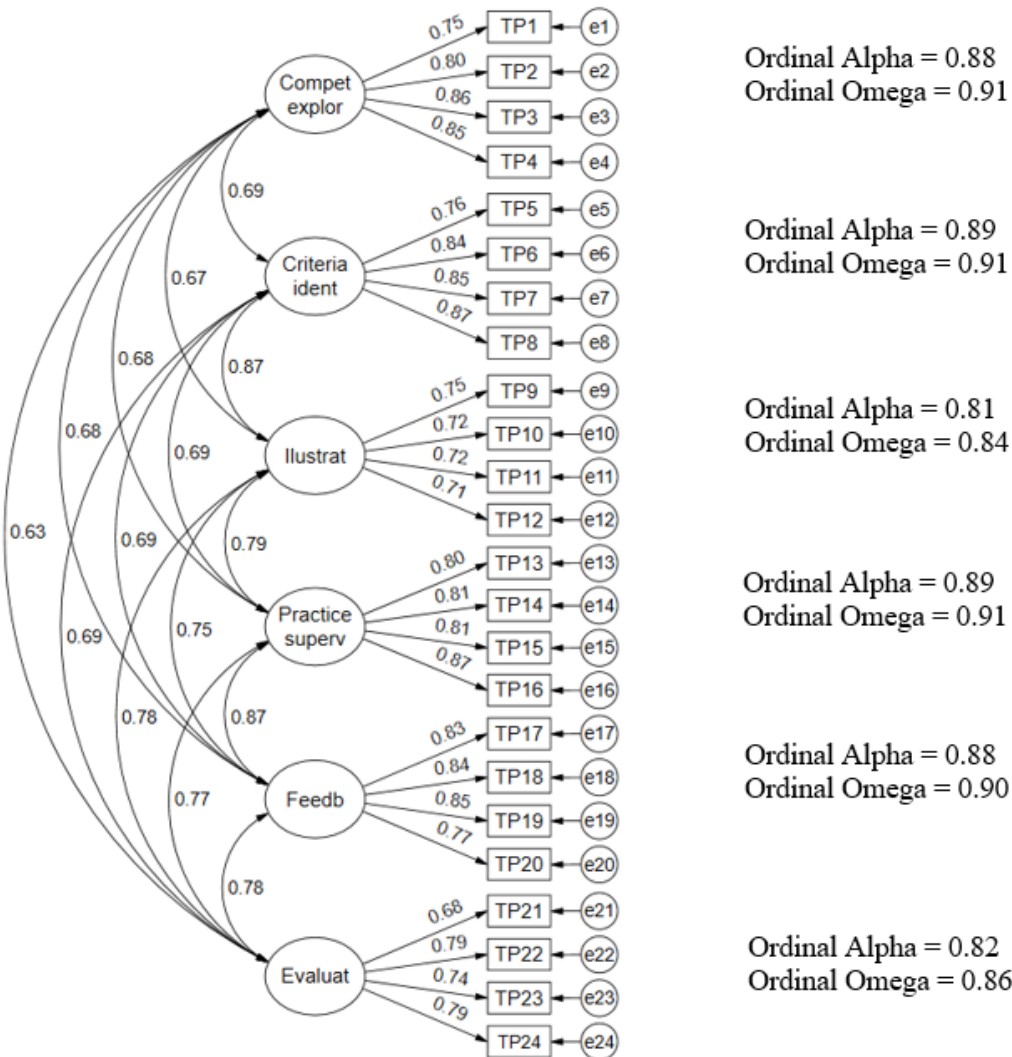

**Figure 1.** Internal structure of the multifactorial model of teaching performance.

**Scale of student self-evaluation of their didactic performance in virtual classes.**

To identify the criteria (variables) of student didactic performance in didactic interactions in virtual classes and practices, as perceived by students of biological sciences at a public university in Peru, the Student Didactic Performance scale, validated with Peruvian graduate students in educational sciences [6,29], was adapted from the Didactic Interactions questionnaire (CID, in Spanish) and validated with Mexican pre-university students in the area of natural sciences [7]. This 24 question scale is organized into 6 student didactic performance criteria, with 4 items for each performance criterion (Appendix B). The 6 student performance criteria and the 24 items correspond strictly to the 6 criteria and the 24 items of the student rating scale on teaching performance in virtual classes. Each student response was rated on a Likert scale, 0 = never, 1 = sometimes, 2 = almost always, and 3 = always.

Figure 2 presents the factor structure of the Student Self-Rating scale on their didactic performance in virtual classes. According to the AFC goodness-of-fit indices, the multidimensional model presents satisfactory evidence of validity based on the internal structure of the construct. The overall fit results were all optimal: $\chi^2$ (234) = 635.140, CFI = 0.954, TLI = 0.946, RMSEA = 0.072 (0.065, 0.079), SRMR = 0.062. With the exception of the illustration—participation factor, all factors showed standardized factor loadings higher than 0.70.

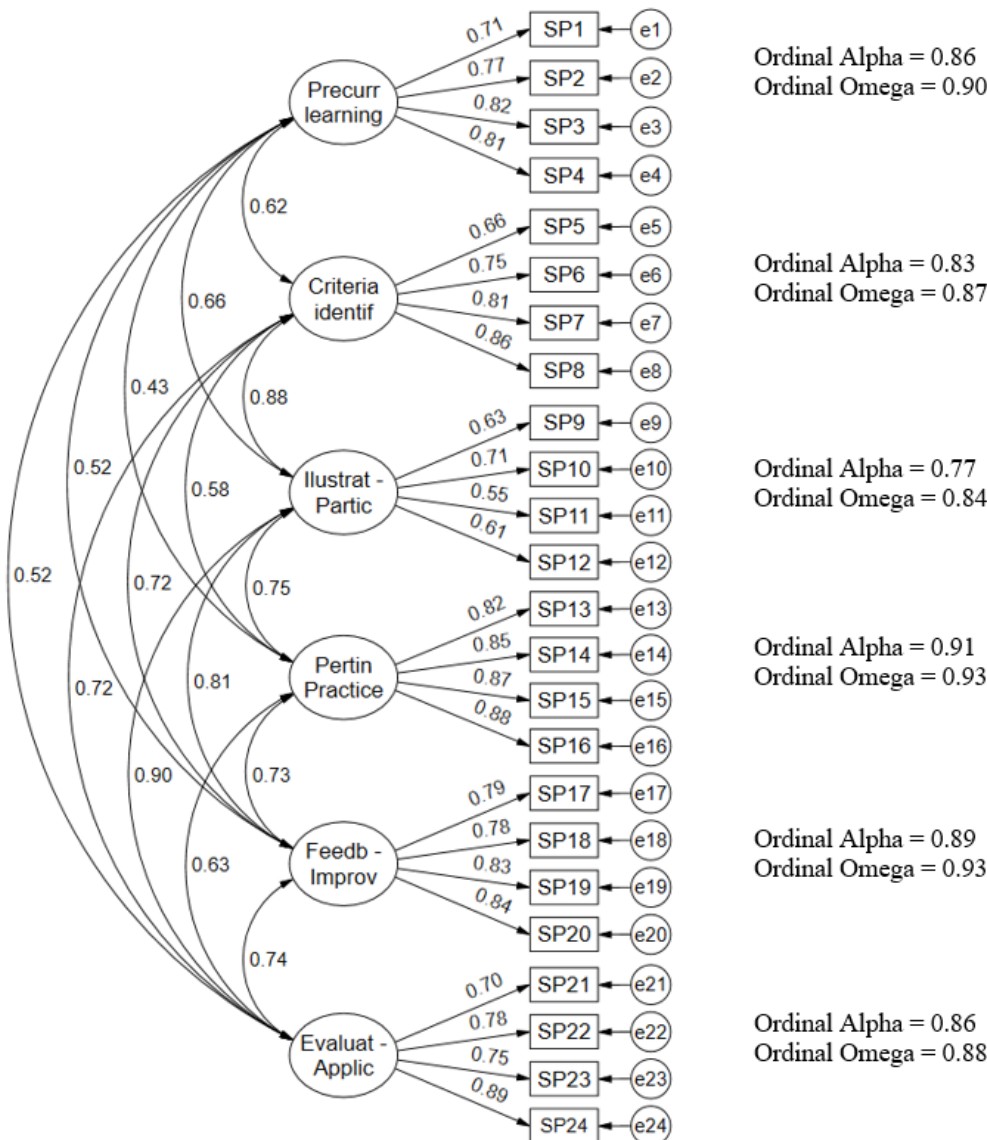

**Figure 2.** Internal structure of the multifactorial model of teaching performance in virtual classrooms.

*2.6. Ethical Principles*

The protocol of this study was reviewed and registered by the Research and Innovation Unit and approved by the Faculty of Biological Sciences of the Universidad Nacional de San Cristóbal de Huamanga. The research was conducted in accordance with the Declaration of Helsinki. In addition, there was a guarantee of no coercion of the participants at the time of recruitment and/or at the time of signing the informed consent. The invitation was sent by e-mail, emphasizing the voluntary nature and disinterested participation of the students, and they were informed that there would be no negative consequences if they chose not to participate in the study. Subsequently, the participants signed a virtual informed consent form.

*2.7. Data Collection*

An invitation to participate in the study was sent to all students enrolled in the 2023-II cycle via the university's institutional e-mails, as well as information on the research, its objectives, benefits, duration, etc. (See Annex 3). Both scales were converted to Google Forms format and permission to access the forms in order to answer the scales was distributed by email to those students who confirmed their desire to participate in the study. The Google Forms format includes the information and informed consent to be completed by the students, and only after that was the link opened in order for the students to respond to the self-report scales on didactic performance.

We received online responses from 330 students; a total of 330 teacher performance questionnaires and 330 student performance questionnaires were collected. The application was conducted over ten days in the month of November 2020. All questionnaires were fully completed, so none of the completed scales were excluded for the realization of the database and subsequent statistical analyses.

*2.8. Data Analysis*

Using the strategy of structural equation modeling, the evidence of validity based on the internal structure of the construct was analyzed with both confirmatory factor analysis (CFA) and covariance analyses of teacher–student performance constructs. Given the categorical nature of the items, the measurement models and structural relationship models were estimated using diagonally weighted least squares with mean and variance corrected (WLSMV) and the models were evaluated with typically recommended goodness-of-fit statistics such as the comparative fit index (CFI), the Tucker Lewis index (TLI), root mean square error of approximation (RMSEA) and standardized root mean square residual (SRMR). CFI and TLI indices $\geq 0.90$ indicate adequate fit and $\geq 0.95$ good fit; likewise, for RMSEA and SRMR, indices $\leq 0.08$ denote adequate fit and values $\leq 0.05$ good fit. The statistical programs used were SPSS version 25 for Windows and the R program version 4.0.2 with Lavaan 0.6–7 and semTools 0.5–3.

## 3. Results

*3.1. Relationships between Teacher Performance and Student Performance in Virtual Classrooms*

A theoretical model that postulates a functional correspondence between the didactic performance of the teacher and the didactic performance of the student, despite the fact that these are measured with self-reports on classes or practices that have already been completed, must empirically test the covariation relationships (association) between constructs that are derived from such a substantive theory of performance measurement in classes and practices, in this case, in the virtual modality. The model obtained with structural equations, which is presented in Figure 3, shows precisely how the didactic performance of the teaching staff is associated (significantly and with a covariation coefficient = 0.83) with the didactic performance of the students in the biological sciences undergraduate program.

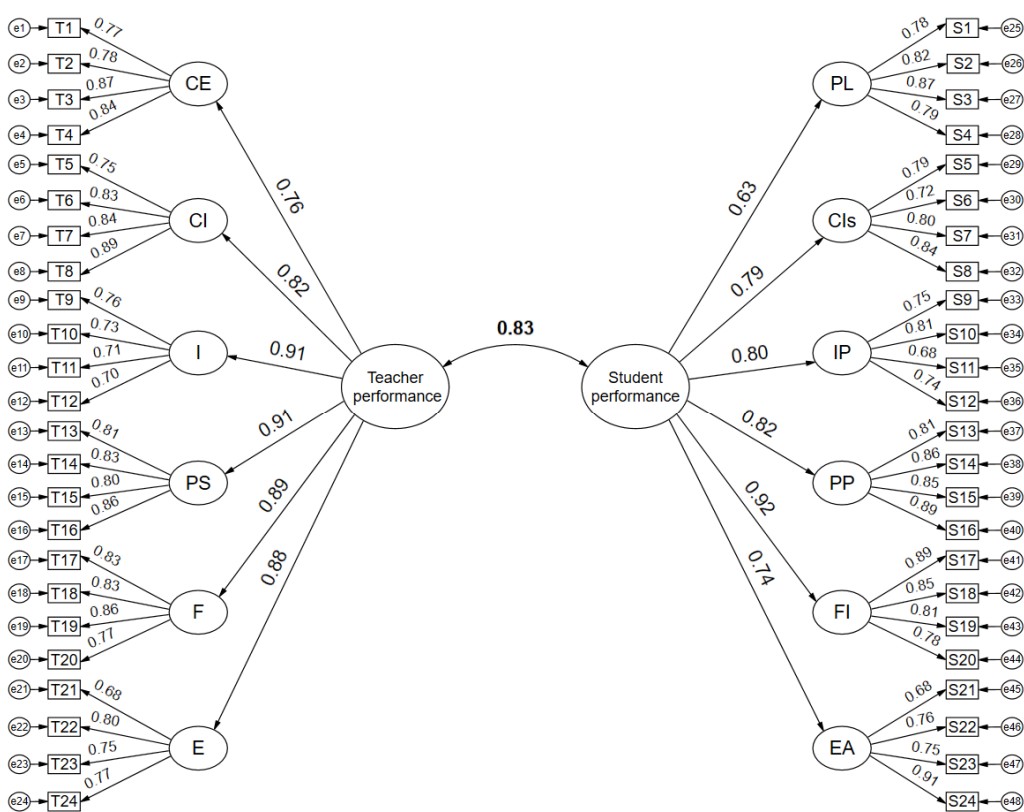

**Figure 3.** Divergence and convergence model between teacher performance and student performance. Note: CE = competency exploration, CI = criteria explicitness, I = illustration, PS = practice supervision, F = feedback, E= assessment, PL = precurrent learning, Cis = criteria identification, IP = illustration—participation, PP = pertinent practice, FI = feedback—improvement, EA = assessment—application.

Both constructs—teacher performance and student performance—are second-order latent variables. Each second-order latent variable shows convergent construct validity with its respective first-order latent variables (the six didactic performance criteria within each second-order variable), reaching regression coefficients between 0.63 and 0.91. Likewise, this resulting model obtained good global goodness-of-fit indices for the divergent model: $\chi^2$ (1067) = 2065.69, CFI = 0.94, TLI = 0.94, RMSEA = 0.05 (0.050, 0.057), SRMR = 0.07.

Additionally, Table 3 presents the latent correlations with which to examine the convergence between factors of the same construct and the divergence between two different constructs (teacher performance versus student performance). It is clearly observed that the correlations between the factors of the same construct are much higher than between the correlations of one construct with the other. The correlations in black are lower than the correlations in blue and red, as shown in the Table 3. The colored values denote convergence, while the black data show divergence.

**Table 3.** Matrix of latent correlations between the factors of teacher performance and student performance.

| | 1 | 2 | 3 | 4 | 5 | 6 | a | b | c | d | e |
|---|---|---|---|---|---|---|---|---|---|---|---|
| **1.** Ex. Competence | 1.00 | | | | | | | | | | |
| **2.** Id. Criteria | 0.62 | 1.00 | | | | | | | | | |
| **3.** Illustration | 0.69 | 0.74 | 1.00 | | | | | | | | |
| **4.** Supervision | 0.69 | 0.74 | 0.82 | 1.00 | | | | | | | |
| **5.** Feedback | 0.67 | 0.72 | 0.80 | 0.80 | 1.00 | | | | | | |
| **6.** Evaluation | 0.66 | 0.71 | 0.79 | 0.79 | 0.77 | 1.00 | | | | | |
| **a.** Precurrent | 0.39 | 0.42 | 0.47 | 0.47 | 0.46 | 0.46 | 1.00 | | | | |
| **b.** Criteria | 0.49 | 0.53 | 0.59 | 0.59 | 0.57 | 0.57 | 0.49 | 1.00 | | | |
| **c.** Participation | 0.50 | 0.54 | 0.60 | 0.60 | 0.58 | 0.58 | 0.50 | 0.62 | 1.00 | | |
| **d.** Practice | 0.51 | 0.55 | 0.62 | 0.62 | 0.60 | 0.59 | 0.52 | 0.64 | 0.65 | 1.00 | |
| **e.** Enhancement | 0.58 | 0.62 | 0.69 | 0.69 | 0.67 | 0.67 | 0.58 | 0.72 | 0.73 | 0.75 | 1.00 |
| **f.** Evaluation Appl. | 0.46 | 0.50 | 0.55 | 0.55 | 0.54 | 0.54 | 0.46 | 0.58 | 0.59 | 0.60 | 0.68 |

*3.2. Analysis of Convergence and Divergence Models of Teacher–Student Didactic Performance*

To analyze the functional correspondence of each of the six pairs of didactic teacher–student performance criteria in virtual classes and practices in biological sciences, six convergence and divergence models were tested, in accordance with the substantive theory from which these measurements are derived [6–9,39,40]. Figure 4 shows the obtained model of divergence and convergence between competent exploration for teachers and precurrents for student learning in virtual classrooms. This model shows satisfactory fit indices: $\chi^2$ (19) = 125.508, CFI = 0.967, TLI = 0.952, RMSEA = 0.084 (0.071, 0.098), SRMR = 0.069. Additionally, Figure 4 shows that the covariance between the constructs has a lower value (divergence) than the factorial relationships (convergence), but that it is significant.

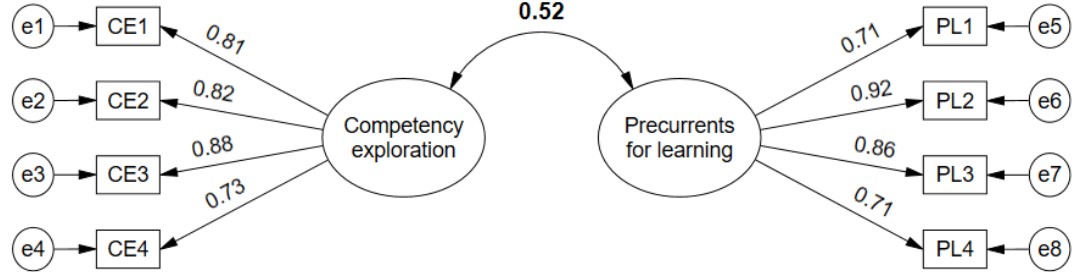

**Figure 4.** Model 1 of competences exploration by teachers—students' percurrent.

Figure 5 shows the divergence and convergence model between explicitness of teacher criteria and identification of student criteria in virtual classes, finding a significant covariation, but less so in comparison with the factor loadings; however, the presence of divergence and convergence is verified for both. The indicated model presents indexes that denote the existence of a very good fit: $\chi^2$ (19) = 51.327, CFI = 0.990, TLI = 0.985, RMSEA = 0.047 (0.031, 0.063), SRMR = 0.045.

Figure 6 presents the divergence and convergence model between enlightenment and enlightenment—student engagement for teachers in virtual classes. According to the fit indices the model is very satisfactory: $\chi^2$ (19) = 51.327, CFI = 0.990, TLI = 0.985, RMSEA = 0.072 (0.048, 0.096), SRMR = 0.045. Additionally, the factor weights on the constructs are higher than the covariance, implying the existence of divergence and convergence.

The data observed in Figure 7 provide support for the validity of the divergence and convergence model between supervision of teacher practice and relevant student practice in virtual classrooms because the overall fit indices of the model are excellent: $\chi^2$ (19) = 44.93, CFI = 0.994, TLI = 0.991, RMSEA = 0.029 (0.000, 0.041), SRMR = 0.029. In addition, it we are

able to see, in a satisfactory manner, that the covariance (0.72) is less than the relationships that the constructs maintain with their respective manifest variables (≥0.79).

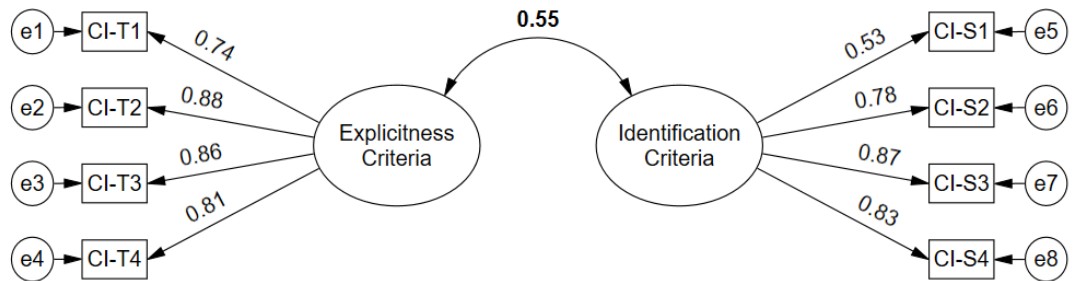

**Figure 5.** Model 2 on explicitness of teacher criteria—identification of student criteria.

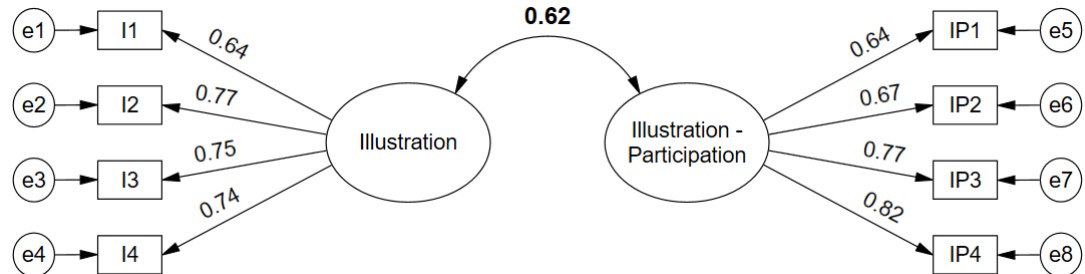

**Figure 6.** Divergence and convergence model 3 on teacher–student teaching performance.

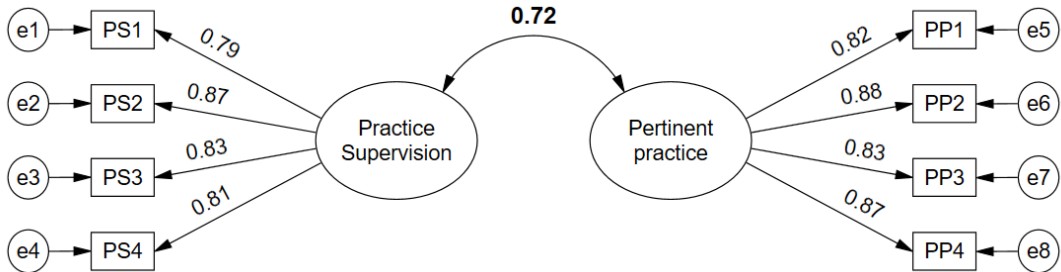

**Figure 7.** Divergence and convergence model 4 on teacher–student didactic performance.

Figure 8 analyzes the divergence and convergence model between feedback and feedback—student improvement for teachers in virtual classes. The estimated indices for the model assessment denote a satisfactory fit: $\chi^2$ (19) = 125.939, CFI = 0.972, TLI = 0.958, RMSEA = 0.070 (0.058, 0.082), SRMR = 0.059. We can also see, in Figure 8, that the covariance, despite being lower than the factorial saturations, denotes the presence of divergence and convergence.

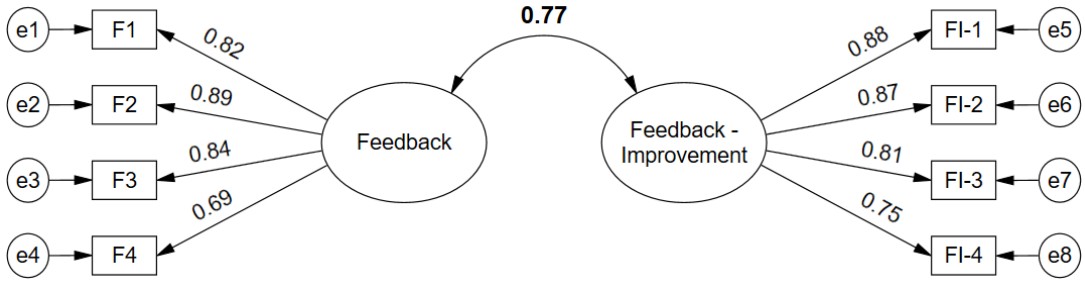

**Figure 8.** Divergence and convergence model 5 on teacher–student didactic performance.

Figure 9 shows the resulting model of divergence and convergence between evaluation and evaluation—student application for teachers in virtual classes, with indices that were very satisfactory: $\chi^2$ (19) = 45.487, CFI = 0.987, TLI = 0.980, RMSEA = 0.045 (0.029, 0.062), SRMR = 0.059. Likewise, the covariance index between both performances was of lower value than the relationships between the constructs with their respective manifest variables; however, the existence of divergence and convergence between both constructs is corroborated.

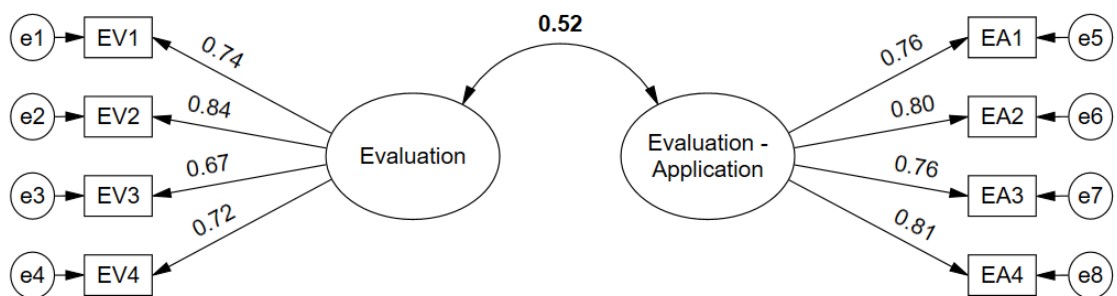

**Figure 9.** Divergence and convergence model 6 on teacher–student teaching performance.

*3.3. Descriptive Analyses of the Factors of Teacher Performance and Student Performance*

All of the descriptive data shown in Table 4, and in particular the Z-score, show that the factors "explicitness of criteria" and "evaluation" are where teachers present the best performance according to the perception of biological sciences students. In contrast, the factors "competency exploration" and "feedback" stand out among those with the lowest performance.

**Table 4.** Descriptive statistics of student-perceived teacher performance factors.

| | Percentile | | | Md | M [95% CI] | SD | Z |
|---|---|---|---|---|---|---|---|
| | 25 | 50 | 75 | | | | |
| Competence exploration | 3 | 5 | 7 | 5 | 5.09 [4.72, 5.46] | 2.65 | −0.44 |
| Explanation of criteria | 5 | 7 | 8 | 7 | 6.82 [6.47, 7.17] | 2.49 | 0.26 |
| Illustration | 5 | 6 | 8 | 6 | 6.53 [6.21, 6.84] | 2.26 | 0.14 |
| Practice supervision | 4 | 6 | 8 | 6 | 6.23 [5.88, 6.58] | 2.51 | 0.02 |
| Feedback | 4 | 5 | 7 | 5 | 5.69 [5.33, 6.05] | 2.55 | −0.20 |
| Evaluation | 5 | 7 | 8 | 7 | 6.74 [6.40, 7.07] | 2.39 | 0.23 |

According to the data in Table 5, the factors referred to as those with the highest compliance in student performance correspond to "identification of criteria" (Z = 0.41) and "evaluation—application" (Z = 0.27). On the other hand, among the factors with the lowest performance are "precurrents for learning", "illustration—participation" and "feedback—improvement".

**Table 5.** Descriptive statistics of student performance factors perceived by students.

| | Percentile | | | Md | M [95% CI] | SD | Z |
|---|---|---|---|---|---|---|---|
| | 25 | 50 | 75 | | | | |
| Precurrent for learning | 4 | 5 | 7 | 5 | 5.39 [5.13, 5.66] | 2.44 | −0.55 |
| Identification of criteria | 6 | 8 | 9 | 8 | 7.59 [7.36, 7.82] | 2.13 | 0.41 |
| Illustration—Participation | 5 | 6 | 8 | 6 | 6.26 [6.04, 6.48] | 2.05 | −0.17 |
| Relevant practice | 5 | 7 | 8 | 7 | 6.82 [6.54, 7.09] | 2.57 | 0.07 |
| Feedback—Improvement | 5 | 7 | 8 | 7 | 6.56 [6.29, 6.82] | 2.44 | −0.04 |
| Evaluation—Application | 6 | 8 | 8 | 8 | 7.27 [7.04, 7.51] | 2.17 | 0.27 |

## 4. Discussion

The first aspect to point out is that the results obtained, first, confirmed the existence of a high (0.83) and significant covariation between constructs of teacher didactic performance and student didactic performance in virtual classes based on the self-reports of biological sciences students from a public university in Peru during the SARS-CoV-2 pandemic. Evidence of divergent and convergent validity of teacher–student didactic performance constructs (two second-order latent variables) and their respective six didactic performance criteria (first-degree latent variables) was also found.

The significant covariation between these two constructs of teaching performance implies, theoretically, a functional correspondence between teacher–student teaching performances. That is to say that the student's behavior is functionally adjusted to the teacher's behavior [8], in the domains of teaching performance proposed in the interbehavioral model of didactic performance [9,39,40], and is consistent with the conception of an interactive field that configures psychological and pedagogical interactions [35–37], at various hierarchically organized levels [38].

The convergent and divergent construct validity results of the confirmatory factor analysis with structural equation modeling presented in the instruments section had previously confirmed the robustness of these two measures in accordance with the fundamentals of a measurement of psychological and educational variables [43–45]. In the case of the measurement of these two types of didactic performance, previous studies have shown that it has been consistent and stable in terms of its construct and content validity, at different educational levels, different disciplines, different contexts and in both face-to-face and virtual modalities of perceived didactic interactions [6,7,29,41]. In other words, in addition to giving empirical support to the interbehavioral model of didactic performance in accordance with the underlying substantive theory [43], the results of this study confirm once again, and in the context of teaching in the training of Peruvian professionals in biological sciences, the conceptual soundness of the constructs and indicators evaluated via self-reporting on teacher and student performance in the virtual classes that were imposed by the pandemic.

As a second point to highlight, this research also sought to validate the convergence and divergence in each of the six pairs of teacher–student didactic performance criteria, as perceived by the student body. The results show the correspondence between each teacher didactic performance competency with its student didactic performance pair. This confirms the assumption that, if the teacher implements didactic experiences focused on interaction in face-to-face or virtual classes [1–3], the students develop their competencies and skills in correspondence with the expected learning and the criteria established by the teacher for the achievement of such expected learning [4,5], developments which are achieved through didactic interactions. Additionally, these results confirm the functional correspondence of the student's behaviors (in six didactic performance criteria), with the didactic behaviors of the teacher (in six didactic performance criteria).

Our results are congruent with the strategies for measuring didactic interactions that complement different approaches to evaluate the quality of instruction [11,13–15] and teacher performance in natural classroom interaction scenarios [12]. In this sense, although it is true that the use of self-reports has been widespread when evaluating teacher performance, teacher behaviors have been traditionally assessed in isolation or independently from the assessment of student didactic performance, despite the fact that they relate to the same interactive situations in which the teacher is said to deploy the assessed teaching behaviors. In this study, our results show that, in each of the six teacher–student didactic performance categories, the level of convergence and divergence of the performance criteria constructs is congruent with the theoretical model of didactic performance assumed in this study [38–40]. This will make it possible to evaluate the differential structuring for each pair of didactic performance criteria, and thus develop differential strategies with which to improve the interactive processes in the teaching of various disciplines in the university context [8,9,39,40].

The third point to highlight is the relevance of these measurements when assessing didactic performance in the virtual classroom interactions that were imposed due to the SARS-CoV-2 pandemic for programs that were originally and exclusively taught face-to-face. Given the rapid adaptation of undergraduate and graduate education institutions in order to implement pandemic-imposed virtual programs, and the innovations they made to the teaching–learning contexts [16–23], it was necessary to characterize and analyze these didactic interactions in a new environment [24]. In that sense, this research also allowed the characterization of the nature of didactic interactions, as well as the teacher's and the student's own performances as perceived by the student body, under the underlying theoretical model. The methodological strategy employed converges with previously conducted studies in its focus on an analysis of the performances related to didactic interactions in the virtual modality [25–32].

When characterizing the didactic performances in classes or practices in the virtual modality, according to the perception of the students in the biological sciences, a greater presence of teacher performances was reported in the categories "identification of criteria" and "evaluation". On the other hand, with respect to student performance, "identification of criteria" and "evaluation–application" were identified with greater presence. This finding is important because, in the theoretical and practical virtual learning interactions in biological sciences, the explicitness and identification of expected learning and achievement criteria are more frequent and correspond to the high valuation of occurrence that the students themselves consider and to the identification of learning criteria by the students. Likewise, the competencies related to the evaluation and application of learning in the virtual modality, both as teacher performance and student performance, are also activities with high frequency in these interactions.

These data seem to indicate that the didactic interactions in the pandemic-imposed virtual modality in the teaching of biological sciences at this university are characterized by performances of specific criteria that determine how to achieve learning with a high frequency of learning evaluation practices. In other words, the students of this discipline of biological sciences value, as performance criteria, those that were most present in the didactic interactions in the virtual modality.

On the other hand, in regard to the didactic performance of the teacher according to the students, the criteria that occur less frequently are "competency exploration" and "feedback", and in the performance of the students, the least frequent type of performance was the criterion "precurrent for learning". That is to say, the teachers during the virtual classes or practices take less opportunities to evaluate the basic and necessary competencies for the learning of their subject or the topic to be developed. Likewise, the data also suggest that these teachers offer little feedback of the student's learning, explaining why the students may also present examples of precurrent (necessary basic competencies) for learning and of improvement in their learning less often.

These results shed light on how the various didactic performance criteria are presented in virtual classes, according to the perception of the students themselves, and allow us to obtain more knowledge of the development of didactic competencies in a new environment that has brought important changes in our view of teaching and learning in the university context. Our findings coincide with research reports on competencies developed in intervention programs during virtual classrooms during the SARS-CoV-2 pandemic, which have also enhanced the sustainable self-development of students [33,34].

A fundamental contribution of this study is the empirical validation of the pertinence of constructs derived from a particular theoretical model [35–38] for the analysis of performances on in-person or virtual didactic interactions. This model assumes the functional correspondence between didactic performances of the teacher and didactic performances of the student [8,9,39,40]. Another contribution of this work is the way in which it fills a research gap in such a way that describes the functional correspondence of teacher–student performance criteria with categories that describe behaviors, skills and competencies in didactic interactions. This model had previously demonstrated its theoretical and empirical

pertinence in the identification of the effect of some teacher didactic performances on some student didactic performances in virtual interactions [6] and in describing teacher and student didactic performances [29] in graduate education sciences; however, the functional relationships between each pair of teacher–student didactic performances had been modeled neither in theoretical nor in practical class interactions.

This study will contribute to the development and improvement of didactic practices in the in-person or virtual environment, because it will allow the precise identification, through the two self-report scales, of how each pair of didactic performance criteria are related, from a theoretical model focused on the nature of didactic interaction. This identification will allow the development of actions with which to improve didactic performance in higher education, both in the virtual environment and in-person classes.

The improvement of these didactic competencies allows both teachers and students to strengthen their sustainable and autonomous development in order to perform better both in academic activities and in their work and social life, contributing to the development of better opportunities for learning and to the improvement of the quality of higher education.

External validity may be a weakness of this research, as a non-probabilistic sampling was carried out in a public university and in a professional career; therefore, the possibilities of generalization should be explored with care. Despite this limitation, the results of the present study are valuable to strengthen the few existing studies that analyze the didactic interactions and teaching competencies exercised in order to favor the learning of university students in a virtual environment. Future studies with probabilistic samples and different professional careers are suggested.

## 5. Conclusions

A first conclusion to be derived from the findings of the present research is that the didactic performance of students in virtual classes of the Professional School of Biology of a public university in Peru, is functionally associated with the didactic performance of the teacher, in the context of the SARS-CoV-2 pandemic, in the August–November period (Semester 2020-I). This implies that the design, planning and execution of didactic experiences centered on interaction favors and optimizes the competency performance of the teacher in teaching and, consequently, the achievement of learning in the process of professional and scientific training of university students.

A second conclusion, derived from the first specific objective of this study, is that convergent and divergent validity was achieved between the six pairs of constructs of teacher didactic performance and student didactic performance, during virtual classes in the biological sciences. The teaching didactic performance criteria were significantly related in each of the following six pairs of performance: 1. exploration of competencies and precurrent for learning, 2. explicitness of teacher criteria and identification of student criteria, 3. teacher illustration and illustration—student participation in virtual classes, 4. supervision of the practice by the teacher and pertinent student practice, 5. teacher feedback and feedback—student improvement, and 6. teacher evaluation and evaluation—student application.

Regarding the second specific objective of this research, and as a third conclusion, it can be pointed out that, in didactic interactions in classes or virtual practices during the SARS-CoV-2 pandemic, according to the perception of the students, there is a greater presence of the performance criteria of explicitness and identification of expected learning and achievement criteria. Likewise, a high frequency of competencies related to the evaluation and application of learning in virtual modality were presented.

**Author Contributions:** Conceptualization, A.B.-R., H.A.-A. and W.C.-L.; methodology, A.B.-R., W.C.-L. and M.A.B.-R.; software, W.C.-L.; validation, W.C.-L. and A.B.-R.; formal analysis, H.A.-A.; investigation, A.B.-R., H.A.-A., V.C.-L. and R.B.A.-G.; data curation, A.B.-R., W.C.-L. and M.A.B.-R.; writing—original draft preparation, A.B.-R.; writing—review and editing, A.B.-R., H.A.-A. and W.C.-L.; visualization, M.A.B.-R.; supervision, H.A.-A. and V.C.-L.; project administration, R.B.A.-G. All authors have read and agreed to the published version of the manuscript.

**Funding:** This research received no external funding.

**Institutional Review Board Statement:** The study was conducted in accordance with the Declaration of Helsinki and was reviewed and registered by the Research and Innovation Unit and approved by the Faculty of Biological Sciences of the National University of San Cristobal de Huamanga by means of the Dean's Resolution No. 092-2020-UNSCH-FCB-CF dated 6 November 2020.

**Informed Consent Statement:** Informed consent was obtained from all study participants.

**Data Availability Statement:** The database of this study is available to interested researchers. It is necessary to write to the corresponding author to obtain it.

**Conflicts of Interest:** The authors declare no conflict of interest.

## Appendix A

Student Rating Scale on Teaching Performance in Online Classes
Instructions:
The following is a scale of values for students to develop regarding the frequency in which they attended the virtual classes of a course they have just completed and in which they evaluate the teacher's performance actions.

For each of the following statements, mark one of the four options that best represents the frequency of its occurrence in the course you are evaluating, choosing one of the four options: 1 = Never, 2 = Almost Never, 3 = Almost Always, 4 = Always

Teacher Performance. Competence exploration (F1)

DD1. At the beginning of the present course (academic cycle), the teacher evaluated my previous knowledge, orally and/or by means of a written questionnaire or paper.

DD2. At the beginning of each class, the teacher explored my skills and knowledge of the topic to be developed in the class.

DD3. At the beginning of each learning unit, the teacher assessed my level of mastery, according to the competencies outlined in the syllabus.

DD4. At the beginning of each new topic (learning unit), the teacher asked about concepts related to the topic to be developed.

Teacher Performance. Identification of criteria (F2)

DD5. At the beginning of a learning unit, the teacher explained what is to be the expected achievement in that unit.

DD6. The teacher explained the criteria we had to meet to perform an activity or task.

DD7. The teacher explained the criteria I needed to fulfill to complete an exercise in class, or a practice of the course.

DD8. In each class, the teacher clearly explained the achievement criteria to be met by the student in each class.

Teacher Performance. Illustration (F3)

DD9. The teacher clearly explained the topic of the class.

DD10. The teacher provided examples of how to develop a task or exercise.

DD11. The teacher described how a biology graduate would solve a problem related to his/her specialty.

DD12. The teacher solves problems in real time, based on the topic he/she has developed.

Teacher Performance. Practice supervision (F4)

DD13. During the classes or practices of the course, the teacher supervises my performance.

DD14. The teacher provides guidance and accompaniment during the course activities.

DD15. The teacher establishes the criteria for the practices and monitors that they are carried out effectively.

DD16. In the course activities, the teacher guides our learning and development of competencies.

Teacher Performance. Feedback (F5)

DD17. The teacher guided my performance in class activities, pointing out my strengths and weaknesses.

DD18. The teacher pointed out my mistakes and taught me the correct way to correct them.

DD19. The teacher teaches me different ways in which I can meet the criteria of the class activities.

DD20. The teacher reviews the work I leave, and provides directions for correcting, and improving my work.

Teacher Performance. Evaluation (F6)

DD21. The teacher performs periodic evaluations of my theoretical knowledge.

DD22. The teacher carries out evaluations of an applicative nature and the solution of practical problems derived from the course.

DD23. The teacher evaluates my ability to integrate knowledge from other courses with this course.

DD24. The teacher evaluated the students according to the learning objectives indicated at the beginning of the course, which are found in the syllabus.

## Appendix B

Student Rating Scale of Student Performance in Online Classes

Instructions:

The following is a scale of values to be developed by the students on the frequency in which they attended the virtual classes of a course they have just completed and in which they evaluate the student's own performance actions.

For each of the following statements, mark with the cursor one of the four options that best represents the frequency of its occurrence in the course you are evaluating, choosing one of the four options: 1 = Never, 2 = Almost Never, 3 = Almost Always, 4 = Always.

Precurrent for learning (A1)

DE1. At the beginning of this course, I demonstrated my previous knowledge, according to the teacher's evaluation.

DE2. At the beginning of each class, I responded and showed the teacher my skills and knowledge of the topic to be developed in the class.

DE3. At the beginning of each learning unit, I demonstrated my level of mastery according to the competencies indicated in the syllabus.

DE4. At the beginning of each new topic (learning unit) I answered the teacher's questions about concepts related to the topic to be developed.

Criteria identification (A2)

DE5. At the beginning of a learning unit I identified what was the expected achievement in that unit (the learning objective to be achieved in the course).

DE6. I completed the activity or task, complying with the criteria explained by the teacher.

DE7. When I completed an exercise or practice of the course, I fulfilled the criteria established by the teacher.

DE8. I applied the achievement criteria established by the teacher for my learning in the class.

Illustration—participation (A3)

DE9. I understand without any problem the subject of the class.

DE10. I developed the task or exercise, following the model provided by the professor.

DE11. I solved a problem related to my specialty as a biology graduate would do.

DE12. I solved problems related to the topic developed in real time.

Relevant practice (A4)

DE13. During the classes or practices, I developed the activities under the guidance and supervision of the teacher.

DE14. I developed in undertaking the course activities with the guidance and accompaniment of the teacher.

DE15. I carried out the practices with the monitoring of the teacher and following the criteria he/she pointed out.

DE16. In the course activities, I receive and apply the teacher's guidance to develop my competencies.

Feedback—Improvement (A5)

DE17. My performance in class activities was pertinent to what was expected, because the teacher guided me by pointing out my strengths and weaknesses.

DE18. With the teacher's support, I identified my mistakes and corrected them, improving my performance.

DE19. I carried out the activities using the different forms that the teacher taught me.

DE20. I corrected my work according to the teacher's indications and improved my performance, fulfilling the indicated criteria.

Evaluation—application (A6)

DE21. I demonstrated in the periodic evaluations my mastery of the theoretical knowledge of the subject.

DE22. I developed the applicative evaluations and solved practical problems derived from the course. (DE22).

DE23. I integrated knowledge from other subjects to answer questions of the subject being developed.

DE24. I have acquired knowledge and developed competencies according to the learning objectives indicated in the course syllabus.

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
