# Peer review of "Self-Reporting of Teacher–Student Performance in Virtual Class Interactions in Biological Sciences during the SARS-CoV-2/COVID-19 Pandemic"

_sustainability, doi:10.3390/su152316198_

Round 1
Reviewer 1 Report
Comments and Suggestions for Authors
The theoretical framework is very well founded. It presents a clear and sequential structure.
The references and quotes used are current and impactful.
Section "1.2 The research problem" should be limited to presenting the objective and working hypotheses of the research and not describing the sample, which is more appropriate to do in the following section: 2. Method. This must be fixed.
When talking about the study variables, a table is presented in the form of a Figure (Figure 1), and this figure also presents very poor quality. It looks blurred. This must be corrected.
The results are very well presented. Graphs and tables are greatly appreciated for better understanding. Very timely relationships are established between the different variables that allow the authors to reach very accurate conclusions.
The discussion is very well prepared. The results obtained are related to those of other similar investigations. It is appreciated that the attention to detail has been taken in its writing and the comparisons between investigations.
The conclusions need to be improved. It doesn't look good to present them as a 3-point list. Better if the research objectives are described one by one, whether they have been achieved or not, as well as giving a confirmation or denial response to the working hypotheses. It would also be very good to add some aspects such as the limitations and difficulties that have been encountered in this research, as well as the future lines of research that they will carry out. Therefore, it requires an improvement to fit into the article presented and thus provide it with greater quality.
Author Response
Dear Reviewer, I appreciate your kind comments about our manuscript. We have tried to incorporate and respond to your observations, and in this new version we have highlighted it in red.
- We have corrected and attended to this observation, Section "1.2 The research problem" should be limited to presenting the objective and working hypotheses of the research and not describing the sample, which is more appropriate to do in the following section: 2. Method.
- We have removed Figure 1 and included a table (Table 2) in its place. I hope this improves the quality of presentation of the study variables.
- We have corrected and responded to your recommendation to improve the conclusions, and I hope we have achieved that.
- We have added the limitations and difficulties in the Discussion section (532-538).
Reviewer 2 Report
Comments and Suggestions for Authors
I am privileged to have a chance to review this scholarly piece of work.
The paper reports a study of self-reporting of teacher-student performance in virtual classes interaction during the pandemic period. The introduction gives a sound argument for the research objectives and the variations of didactical context. The research instrument is useful and validated with a clear factor structure. The analysis is carefully carried out and presented. The discussion and conclusions are well supported.
Author Response
Dear Reviewer 2, I thank you for your kind comments on our manuscript.
We have tried to incorporate and respond to all observations and comments, and in this new version we have highlighted it in red.
Reviewer 3 Report
Comments and Suggestions for Authors
The presented text is valuable because of the applied and presented research approach to determine the level of covariance, divergence and convergence between the constructs of teacher's didactic performance and students' didactic performance in virtual classes as a result of the development of the SARS-CoV-2 pandemic by means of confirmatory factor analysis and analysis of covariance of teacher-student performance constructs. The research procedure is described factually and comprehensively legitimizing an interestingly developed theoretical construct.
The elaboration of the theoretical part leaves a deficiency in the characterization of didactic interactions in virtual learning environments during the SARS-CoV-2 pandemic. The authors refer quite often to their own research papers (coinciding in theoretical and research assumptions to the presented text), limiting the review of research reports in the subject area undertaken (61-97).
The research results and study conclusions themselves seem self-evident and do not represent an advance in current knowledge of didactic interactions in the online environment.
The discussion of the results missed deeper references to the possible limitations of the study due to specific needs and capabilities, as well as the psycho-physical condition of the participants of the virtual learning community studied in the context of uncertainty caused by the development of the SARS-CoV-2 pandemic during the study period (447-458).
With regard to the results obtained, there is a lack of concrete proposals for changes in the development and improvement of digital didactical practices and the strengthening of sustainability development (486-492).
Author Response
Dear Reviewer 3, Thank you for your kind comments on our manuscript. We have tried to incorporate and address all observations and comments, and in this new version we have highlighted it in red.
1. Regarding the development of the theoretical part and to overcome some deficiency in the characterization of didactic interactions in virtual learning environments during the SARS-CoV-2 pandemic; we have removed one of the self-citation papers and have included two new references to expand the review of research reports (64 – 101).
2. We have made changes to Discussion and conclusions in order to improve and address the following observation: "The research results and study conclusions themselves seem self-evident and do not represent an advance in current knowledge of didactic interactions in the online environment".
3. In the indicated segment (447-458), we have added references cited in this same manuscript. On the other hand, this study focused mainly and in a general way on the performances of the teacher and the student in interactive processes in virtual classes, considering categories that allow describing the performances of the teacher and the students.
The purpose of this NO was to consider all the factors or dispositional variables that may affect didactic interactions. These variables do not only refer to the physical needs and capabilities, nor only to the psychophysical conditions of the participants. Also the socioeconomic level, the possession of assets at home, the cultural capital of the family of origin, the student's commitment and interest, as well as educational opportunities (Coleman et al., 2016), opportunities for learning (Carroll, 1963; 1989), and the quality of instruction, are variables that, like the abilities and physical conditions of the students, can affect didactic interactions. Precisely in our other publications in journals indexed in SCOPUS we have developed work on these topics, but it is not the purpose of this work. These conditions do not properly correspond to the objectives set out in our manuscript.
4. "With regard to the results obtained, there is a lack of concrete proposals for changes in the development and improvement of digital didactical practices and the strengthening of sustainability development (486-492)".
This study does not specifically focus on improving digital practices, but rather, its findings constitute a fundamental contribution to the empirical validation of the relevance of constructs derived from a particular theoretical model [35-38] for the analysis of performances in face-to-face or virtual didactic interactions. This model assumes the functional correspondence between teaching performances of the teacher and teaching performances of the student [8-9, 39-40].
To address the reviewers' comments, we have added several paragraphs in the discussion (508-531). We have made proposals for change in the development and improvement of educational practices and the strengthening of sustainable development.